# MMCR: ADVANCING VISUAL LANGUAGE MODEL IN MULTIMODAL MULTI-TURN CONTEXTUAL REASONING

## ABSTRACT

Compared to single-turn dialogue, multi-turn dialogue with contextual coherence better aligns with the needs of real-world human-AI interactions. Additionally, as training data, it provides richer contextual reasoning information, thereby guiding the model to achieve better performance. However, existing vision-language models (VLMs) primarily rely on single-turn dialogue training and evaluation benchmarks. In this paper, following the characteristics of human dialogue, such as focused topics and concise, clear content, we present MMCR (Multimodal Multi-turn Contextual Reasoning), a novel dataset comprising: (1) MMCR-310k, the largest multi-turn instruction tuning dataset with 310K contextual dialogues, each covering 1-4 images and 4 or 8 dialogue turns; and (2) MMCR-Bench, a diagnostic benchmark featuring dialogues, spanning 8 domains (Humanities, Natural, Science, Education, etc.) and 40 sub-topics. Extensive evaluations demonstrate that models fine-tuned with MMCR-310k achieve 5.2% higher contextual accuracy on MMCR-Bench, while showing consistent improvements on existing benchmarks (+1.1% on AI2D, +1.2% on MMMU and MMVet). MMCR and prompt engineering will be released publicly.

## 1 INTRODUCTION

The pursuit of establishing Artificial General Intelligence (AGI) capable of providing expert-level responses has long been a pressing goal within the academic community. Such mature agents are expected to engage in multi-turn interactions with human users, process long-context dialogue information, and demonstrate multi-turn instruction following ability, which are also increasingly common requirements in real-world scenarios.

In response to the current state, numerous outstanding studies and benchmarks in the domain of Large Language Models (LLMs) have been developed to enhance their ability to handle long-context multi-turn dialogue scenarios, including but not limited to sparse and efficient attention mechanisms (Dao, 2023; Liu et al., 2022), position encoding extrapolation and dynamic adjustment (Ma et al., 2025), context summarization and memory bank construction (Yu et al., 2025), as well as various multi-turn dialogue fine-tuning datasets and evaluation benchmarks (Bai et al., 2024). For open-source VLMs, the mainstream training and evaluation data (Liu et al., 2024a) are largely confined to handling single-image, single-interaction modes. However, in real-world human-AI interactions, multi-turn dialogues better align with human conversational habits and practical application needs. Furthermore, multi-turn dialogue contain richer contextual reasoning information, which significantly aids in enhancing the performance.

To overcome this problem, some works have attempted to address this issue (Cui et al., 2020; Bai et al., 2024). However, these efforts focus solely on dialogue content. Very recently, MMDU (Liu et al., 2024c) has constructed high-quality training data and benchmarks using GPT-4o (AI, 2024) based on Wikipedia to enhance and evaluate VLMs in such scenarios, but it does not emphasize strengthening contextual logic, maintaining cross-turn dialogue consistency, or parsing long-range contextual dependencies. These capabilities are vital for deploying AGI in real-world settings with continuous multimodal inputs. Logical reasoning over long contexts also improves model performance on public benchmarks. Thus, academia urgently needs a multi-dimensional training and

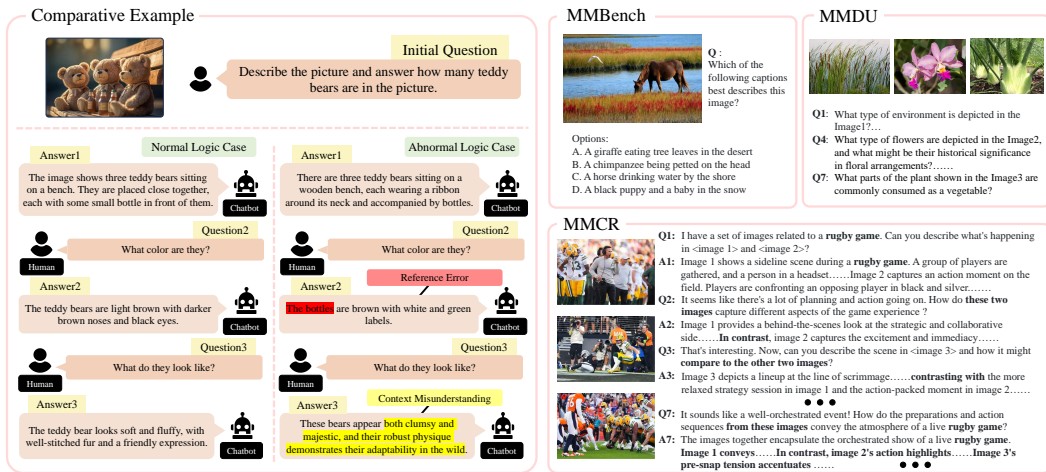

Figure 1: Samples comparison. On the left side: two examples corresponding to a good contextual reasoning sample and a bad contextual reasoning sample with errors during multi-turn dialogue. On the right side: single-image/turn sample from MMBench (Liu et al., 2024a), multi-image/turn sample from MMDU (Liu et al., 2024c), and a sample from our proposed MMCR. We **bold** the content that demonstrates the contextual coherence and thematic consistency inherent in MMCR.

evaluation framework encompassing complex multi-turn dialogues, strong contextual associations, and dynamic scene adaptation.

Building on this, we introduce MMCR (Multimodal Multi-turn Contextual Reasoning), which includes MMCR-310k and MMCR-Bench, a multi-turn instruction fine-tuning dataset with a mix of single/multi-image interactions, and an evaluation benchmark. This instruction fine-tuning dataset and evaluation benchmark are constructed based on multimodal text-image interleaved dataset OmniCorpus Li et al. (2024b), though GPT-4o with the guidance of carefully designed prompt engineering. The generated multi-turn dialogue datasets, which feature strong contextual logic and simulate real-world user-agent interactions, also conduct evaluations on specific dimensions of multi-turn dialogue contextual reasoning.

Specifically, MMCR features the following characteristics: (1) Multi-turn Interaction and Realism: MMCR-310k contains 210k single-image and 100k multi-image multi-turn dialogue data, with dialogues focused on images over four and eight rounds, aiming to simulate real-world user-chatbot interactions. (2) Strong Contextual Relevance and Logical Progression: We emphasize that each round of dialogue should delve into image details, inter-image relationships, and related themes, while maintaining a clear contextual association. (3) Evaluation Openness and Extensiveness: We utilize multi-level annotators provided by Ridnik et al. (2021) to label images from OmniCorpus (Li et al., 2024b), selecting labeled samples from eight major domains and across 40 sub-topics for benchmark. Using GPT-4o as the evaluator, we score the model's responses across five dimensions. This comprehensive evaluation assesses the model's performance in long-context multi-turn dialogues. Several samples from difference datasets are compared in Fig. 1

We validate MMCR using Ovis (Lu et al., 2024b), a state-of-the-art (SOTA) VLM architecture that innovatively employs a learnable visual embedding table for visual encoding. After instruction fine-tuning with MMCR, Ovis achieves obviously improvement on MMCR-Bench, which reveals the limitations of existing models in logical contextual reasoning of multi-turn dialogues and validates the necessity of constructing such a dataset. Besides, incorporating our proposed MMCR into the supervised fine-tuning stage further enhances performance on existing benchmarks. Furthermore, our analysis reveals a subtle but important phenomenon, "Less is More", when training large models: during high-quality supervised fine-tuning, adding more data does not always improve performance. Instead, maintaining balanced data proportions across tasks is crucial; in some cases, reducing data for certain tasks can enhance results, especially for smaller models. This insight is valuable for future supervised fine-tuning of large models.

In summary, the major contributions are:

- We propose MMCR, a multimodal collection featuring a hybrid instruction-tuning dataset (310K dialogues with 4-8 turns) and a GPT-4o-grounded benchmark, specifically designed to enhance and assess the multi-turn dialogue contextual reasoning ability of VLMs in real-world human-AI dialogues.

- Systematic evaluation shows that MMCR enables precise and effective improvements and assessments in the context of multimodal multi-turn dialogues. It also shows enhanced performance on existing benchmarks.

- We discover the "Less is More" phenomenon in supervised fine-tuning and demonstrate it through experiments. This phenomenon emphasizes the importance of data distribution in addition to data volume. It provides valuable insights for future large model training efforts.

## 2 RELATED WORK

### 2.1 MULTI-TURN INSTRUCTION TUNING DATASETS

Recent advancements in instruction tuning (Liu et al., 2023a) have significantly enhanced model capabilities across various tasks. Das et al. (2017) introduced the Visual Dialog task, where AI answers questions about image using dialog history, and released VisDial dataset with 1.2M Q&A pairs to benchmark progress. Alamri et al. (2019) proposed the Scene-Aware Dialog task, requiring models to use video, audio, dialog history, and questions to generate natural responses, supported by the AVSD dataset with 11k annotated videos. Li et al. (2023) introduced a framework that requires minimal annotation to enable LLMs to follow multi-turn multimodal instructions, requiring only image-description pairs to generate multi-turn multimodal instruction-response dialogues from a language model. Niu et al. (2024) introduced a new multimodal, multi-turn posture detection dataset, MmMtCSD, and proposed a novel multimodal large language model framework, MLLM-SD. Maheshwary et al. (2024) proposed a taxonomy-guided, case-based generation of a multilingual, multi-turn instruction fine-tuning dataset, M2Lingual, containing 182K total instruction fine-tuning data. To further align LLM outputs with human, R2S (Hou et al., 2024) used dialogue chain logic to guide LLMs in generating knowledge-intensive multi-turn dialogues for instruction fine-tuning, and constructs K-BENCH and GINSTRUCT instruction datasets using open-source datasets. MMDU (Liu et al., 2024c) clustered related images and textual descriptions from Wikipedia, and constructed question-answer pairs with the help of human annotators using GPT-4o. They proposed the MMDU-45k instruction fine-tuning set and the MMDU evaluation benchmark, paving the way to bridge the gap between current VLM models and practical application needs.

Multi-turn dialogue has attracted considerable interest, resulting in many instruction-following datasets. However, most focus on linguistic aspects within specific domains, with few addressing multi-turn dialogues. Existing datasets often lack logical coherence and clear contextual referencing, resembling collections of single-turn dialogues rather than cohesive, topic-centered multi-turn conversations.

### 2.2 VLM BENCHMARKS

The rapid development of the multimodal field has driven the industry to propose evaluations of the capabilities of the models in various tasks and domains. Nowadays, the academic community has developed numerous high-quality evaluation benchmarks (Lu et al., 2022; Yue et al., 2024; Liu et al., 2023c;b), establishing a standardized and objective evaluation system. However, the evaluation of visual language models has traditionally focused on atomic tasks such as visual question answering (Fu et al., 2024), image understanding and captions (Hiippala et al., 2021), OCR recognition (Liu et al., 2024b), and diagram analysis (Masry et al., 2022), where single-instance understanding is prioritized over contextual coherence. While current benchmarks assess basic perceptual abilities, they fail to capture the contextual coherence and dynamic nature of real-world dialogues. Realistically, context-driven conversations require strong logical consistency, clarity, and a coherent storyline, demanding that models reference dialogue history and integrate iterative knowledge. To address this, we develop MMCR, providing fine-tuning data for multi-turn contextual reasoning in multimodal models. MMCR also evaluates models' ability to handle complex contextual logic, supporting expert-level AGI interactions in real-world scenarios.

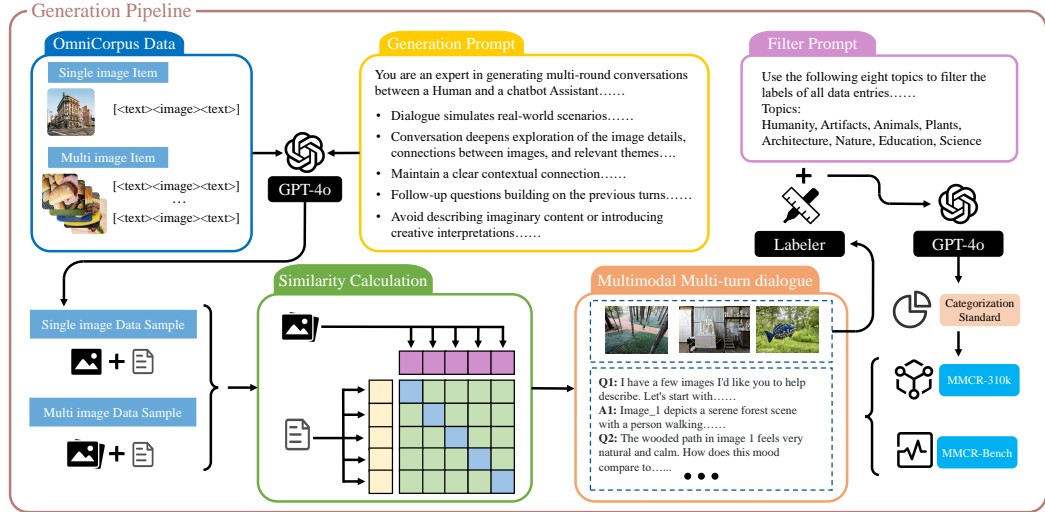

Figure 2: The pipeline of constructing MMCR. First, we randomly select 1.6 million samples from OmniCorpus as base samples. Then, we generate dialogue content using GPT-4o guided by our carefully designed prompt engineering. Next, we filter the constructed dataset using CLIP based on the semantic similarity between images and their corresponding dialogues, resulting in 310k high-quality samples. Finally, we analyze the topics of the obtained samples using an automatic labeler and randomly select samples from 40 topics based on the statistical information to build MMCR-Bench. Please refer to Appendix Section E and Section F for the complete prompt.

## 3 MMCR

### 3.1 OVERVIEW

In both the NLP and multimodal fields, a widely accepted notion is that a longer context window means the ability to receive more extended context, allowing for better information integration and providing more reasonable responses. However, despite significant progress in extending the context window length of VLMs, their practical performance in real-world multi-turn dialogues remains limited, particularly when handling interleaved inputs in long conversation contexts (Liu et al., 2024c).

To further enhance and evaluate the long-context reasoning abilities of VLMs in multimodal, multi-turn dialogue scenarios, we propose MMCR, a dataset that includes a multimodal multi-turn dialogue instruction fine-tuning dataset and corresponding evaluation benchmarks. It consists of MMCR-310k, which includes 210k single-image and 100k multi-image high-quality multi-turn dialogue data with strong contextual logic relationships, and MMCR-Bench, an evaluation benchmark containing 600 elaborate single/multi-image mixed evaluation data with multi-level annotations. The complete construction pipeline is shown in Figure 2. We provide detailed descriptions of MMCR-310k and MMCR-Bench in this section.

### 3.2 MMCR-310K

**Data Collection.** MMCR aims to provide VLMs with multi-turn multimodal dialogue data that has strong contextual relevance and logical reasoning, enhancing and evaluating the performance of existing models in real-world human-AGI interaction scenarios. To construct multi-turn multimodal dialogues with coherent contextual logic, ensuring that the images and text are highly consistent is essential. OmniCorpus (Li et al., 2024b) is a massive multimodal dataset containing billions of image-text pairs, with its open-source version, OmniCorpus-CC-210M, covering a wide range of languages and scenarios from simple to complex. The image-text interleaved data it generates aligns with our goal of high homogeneity and strong relevance between images and text. Based on this, we sample data from OmniCorpus as the foundation for our dataset.

**Generation Pipeline.** We carefully craft prompts to guide GPT-4o in generating multi-turn dialogue data for MMCR based on the provided data. We conduct complex prompt engineering to ensure the

Table 1: Statistics of MMCR-310k and MMCR-Bench.

| Statistic | Number | |
|---|---|---|
| **MMCR-310k** | 210k single | 100k multi |
| - Avg./Max. Image&Text Tokens | 1.4k/1.8k | 3.5k/6k |
| -Images number | 1 | 2 / 3 / 4 |
| -Proportion | - | 61.5%/27.6%/10.9% |
| -Turns | 4 | 8 |
| -Total image | 210k | 270k |
| -Number of QA Pairs | 840k | 870k |
| **MMCR-Bench** | 400 | 200 |
| -Proportion | - | 50.5%/30.5%/19% |
| -Total image | 400 | 537 |
| -Number of QA Pairs | 1.6k | 1.6k |

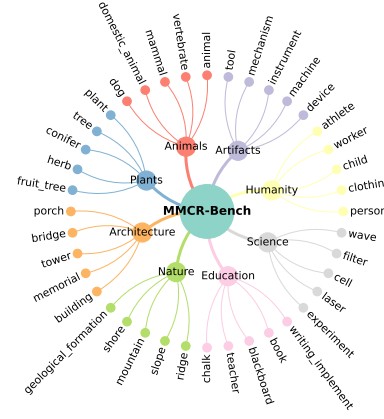

Figure 3: Topic distribution of MMDU-Bench.

generation of multi-turn dialogue data with strong contextual logic. The key focus is: "Ensure that each turn of the dialogue progressively deepens the exploration of image details, connections between images, and related topics. The dialogue must maintain clear contextual continuity, with subsequent questions based on previous requests or questions and their corresponding answers. At the same time, the AI assistant's responses must be detailed and contextually relevant, focusing only on observable details directly inferred from the images, avoiding descriptions of fictional content or introducing creative interpretations." At the same time, we stipulate that the special token marking for each image only appears once at the beginning of the first dialogue where the image is mentioned, such as "Image_n: ⟨image⟩," where n represents the sequence number of the image in the current case. We extract 1.2M single-image samples and 400k multi-image samples from OmniCorpus-CC-210M and combined them with fine-grained prompts to deliver to GPT-4o for multi-turn dialogue data generation.

The resulting samples are then strictly filtered. We use the classic image-text encoder, CLIP (Radford et al., 2021), as a referee, comparing the semantic similarity between images and their corresponding dialogues in the generated results. Using strict regular functions, we filter out entries with missing images, formatting errors, unclear text, and mismatches between images and text, ultimately obtaining 210K single-image samples and 100K multi-image samples.

**Data Statistics.** The proposed MMCR dataset contains a total of 310k single/multi-image mixed multi-turn instruction fine-tuning dialogue data. Tab. 1 shows the specific distribution of our data. Among them, the number of dialogue turns for single-image data is 4, with an average image-text mixed token length of 1.4k and a maximum of 1.8k. For multi-image data, the number of dialogue turns is 8, with an average image-text mixed token length of 3.5k and a maximum of 6k. Each entry in the multi-image data contains 2 to 4 images, and the distribution of different image quantities is 61.5%/27.6%/10.9%. All data follows a strict alternating human-AI assistant dialogue mode, with each dialogue turn closely related to the image. It is important to emphasize that MMCR focuses on presenting multi-turn multimodal dialogues with strong contextual relevance, clear logic, and themes. Therefore, designing data with a large number of images or extending the length of responses in dialogues is not our main goal. Please refer to Appendix Section B for detailed implementation.

### 3.3 MMCR-BENCHMARK

**Data Annotation and Construction.** To conduct a comprehensive evaluation, we aim for the constructed evaluation benchmark to cover as many fields and topics as possible, in order to assess the VLMs' overall knowledge base and comprehension abilities. Therefore, we annotate the data based on the images and set up multiple topics for screening to build an evaluation benchmark with evenly distributed categories and broad coverage, namely MMCR-Bench. We show the distribution of these topics in Fig. 3.

We first use the labeler provided by Ridnik et al. (2021) to annotate the images in the data samples. The pre-trained model of this labeler has been efficiently trained on ImageNet-21k and possesses robust labeling capabilities. The labeler can provide multi-level tags, so we aggregate all the labels

**Evaluation Pipeline**

**Eval Prompt**

### Role Setting

You are an assistant skilled at evaluating the performance of multi-modal models in multi-turn dialogues……

### Scoring rules:

- Precision and Conciseness of Image Descriptions
- Consistency of Contextual References
- Logical Contextual Relationship
    Scores 1-2: The dialogue shows a lack of logical flow with disjointed or disconnected turns.
    ......
    Scores 9-10: The dialogue is flawlessly logical, with seamless and well-reasoned transitions throughout.
- Clarity of Dialogue Theme
- Absence of Redundancy
- Overall Score

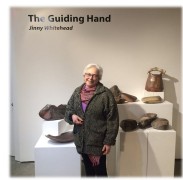

**VLM Response & Reference**

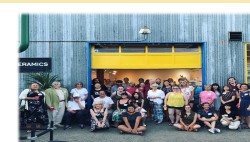

**VLM Response:**
Image 1 shows an art gallery setting with a display titled "The Guiding Hand" by Jinny Whitehead.…… Image 2 features a large group of people gathered outside a venue with a sign labeled……

**Reference:**
In image 1, a woman is standing in front of a wall with the text "The Guiding Hand" and "Jinny Whitehead" written on it…… Image 2 shows a group of people posing for a photo in front of a building with ……

**GPT-4o Judgement**

The assistant's answer met the basic requirements, but there were some overlooked details in the image……

**Scores**:
{'Precision': 6, 'Consistency': 6, 'Logical': 8, 'Clarity': 8, 'Redundancy': 7, 'Overall Score': 7}

Figure 4: MMCR Evaluation Diagram. We provide a prompt, the model's response, and reference, and then let GPT-4o serve as an impartial judge to score the outputs.

from the entire dataset and count the frequency of each tag across all images, before passing the results to GPT-4o for further screening. We have defined a total of eight major categories: humanity, artifacts, animals, plants, architecture, nature, education, and science, and we use GPT-4o to partition the aggregated tag statistics according to these categories. For each major category, the top five most frequent tags are selected as subtopics, resulting in a final set of 40 tags with the highest occurrence across these categories that serve as the coverage categories for MMCR-Bench. Subsequently, data entries that include images with the corresponding tags are considered to belong to the respective topics. Within each major category, we randomly select 10 samples from the single-image data and 5 from the multi-image data, ultimately obtaining a total of 600 multimodal multi-turn dialogue entries to serve as the final MMCR-Bench evaluation benchmark.

**Evaluation.** Using a powerful LLM as an impartial judge to evaluate model responses has always been a common paradigm in both the NLP and VLM fields (Zheng et al., 2023). Inspired by MMDU, we have developed an evaluation pipeline using GPT-4o to assess model performance. We combine the model's inference results on the proposed MMCR-Bench with specific prompts and the corresponding reference answers from the benchmark, then submit them to GPT-4o to score the model's performance. Fig. 4 illustrates the prompt designed for contextual logic reasoning along with the corresponding question and response structure. Unlike MMDU, which emphasizes creativity and richness, our goal is to enhance and assess the contextual reasoning abilities of VLMs in multi-turn dialogue scenarios. Therefore, we place greater emphasis on referential consistency, logical coherence, and dialogues that revolve around a clear topic. We have established five evaluation criteria: precision and conciseness of image descriptions, consistency of contextual references, logical contextual relationships, clarity of dialogue topics, degree of redundancy, and over all score. We guide GPT-4o to score across these dimensions using prompts that divide the range from 0 to 10 into two-point intervals corresponding to different scoring rules. We then aggregate and average the model's scores on the entire MMCR-Bench to obtain a final score.

**Automatic and Manual Quality Control.** To ensure the integrity and reliability of MMCR, we implement two rigorous quality control procedures: (1) We use CLIP to filter out data samples with low similarity between text and corresponding images, ensuring that only high-quality, relevant image-text pairs are retained. (2) We introduce a multi-expert human supervision process in which six experts independently and iteratively review the entire MMCR-Bench, evaluating each item against the five specified criteria (e.g., precision and conciseness (PC)). If any expert identifies a suboptimal data item, it is regenerated and re-evaluated. Items failing to pass after more than three rounds of review are discarded and replaced with new samples randomly selected according to the

Table 2: Evaluation of Ovis on MMCR-Bench. We report the metrics of precision and conciseness (PC) of image descriptions , consistency of contextual (CC) references, logical contextual (LC) relationships, clarity of dialogue topics (CT), degree of redundancy (DR), and overall score (OA). Each metric is magnified tenfold for more precise representation. Ovis-1/4/8B employ Qwen2.5-0.5/3/7B-Instruct (Qwen et al., 2024) as LLM and aimv2 (Fini et al., 2024a) as visual encoder.

| Model | MMCR | PC | CC | LC | CT | DR | OA |
|-------|------|-----|-----|-----|-----|-----|-----|
| Ovis-1B | ✗ | 57.9 | 65.9 | 58.4 | 68.8 | 61.8 | 60.6 |
|         | ✔ | 61.3 *(+3.4)* | 68.4 *(+2.5)* | 62.0 *(+3.6)* | 71.2 *(+2.4)* | 69.3 *(+7.5)* | 65.8 *(+5.2)* |
| Ovis-4B | ✗ | 62.6 | 72.8 | 67.3 | 75.7 | 68.9 | 68.5 |
|         | ✔ | 65.8 *(+3.2)* | 74.6 *(+1.8)* | 69.9 *(+2.6)* | 76.8 *(+1.1)* | 75.0 *(+6.1)* | 72.0 *(+3.5)* |
| Ovis-8B | ✗ | 63.2 | 75.1 | 68.7 | 77.1 | 72.7 | 70.4 |
|         | ✔ | 67.0 *(+3.8)* | 76.5 *(+1.4)* | 72.5 *(+3.8)* | 79.1 *(+2.0)* | 76.6 *(+3.9)* | 74.6 *(+4.2)* |

relevant subtopic. This comprehensive human inspection guaranteed that each text is contextually coherent and highly relevant to image, maintaining both contextual logic and semantic alignment.

## 3.4 CREATION AND EVALUATION OF SYNTHETIC DATA

Beyond 4o, other strong VLMs, even open sourced large scale VLMs can also serve as sources for creation and evaluation. Considering efficiency and cost, we select 4o as creator and judge. Foundational models like 4o indeed cannot perfectly fulfill the roles of human experts in creation and evaluation, but they remain the best choice when considering all factors. As Google DeepMind claimed in Ruibo et al. (2024) that AI synthetic data has proven to be an effective and relatively low-cost alternative in various reasoning tasks. UC Berkeley team (Lianmin et al., 2023) noted strong LLM judges can align with human preferences effectively, both are about 80%. Additionally, we have invested considerable effort in prompt engineering for iteratively optimizing creation and evaluation prompts based on the model's responses to refine their quality. Experimental results also support the efficacy of this data creation and evaluation pipeline.

## 4 EXPERIMENTS

In this section, we present the effectiveness of MMCR on both MMCR-Bench and public evaluation benchmarks. We display training settings in Appendix Section B.

### 4.1 EVALUATION OF MULTI-TURN DIALOGUE CONTEXTUAL REASONING

Tab. 2 presents the results on MMCR-Bench from targeted dimensions. We employ three versions (1B, 4B, and 8B) of Ovis (Lu et al., 2024b), a novel VLM architecture designed to align visual and textual embeddings via a visual embedding table, as VLMs for our experiments. Performance comparison with other open-source and closed-source models in Tab. 3 confirms its SOTA performance. According to the experimental results, we can draw several observations:

- Challenging Evaluation Environment: Even the current leading SOTA models face significant challenges under the MMCR-Bench evaluation framework. For example, Ovis-4B, as the current top open-source model, achieves results predominantly around 70%, indicating that there is still considerable room for improvement in the contextual reasoning abilities of current VLMs.

- Quality of the MMCR Fine-Tuning Dataset: The MMCR instruction fine-tuning dataset yields average improvements of 4.1%, 3.1%, and 3.2% across six evaluation metrics for the three groups of VLMs, demonstrating its excellent quality. Notably, the Ovis-1B model shows an overall improvement of 5.2%, which further validates the contribution and beneficial impact of the proposed dataset on enhancing the contextual reasoning capabilities of multimodal models.

- Mitigating Redundancy in Smaller Models: Interestingly, under the Absence of Redundancy evaluation metric, we observe that smaller models exhibit greater improvements after instruction fine-tuning with MMCR. This suggests that redundancy issues are more

Table 3: Evaluation results for multiple models on various public benchmarks. We use the scores from AI2D (Hiippala et al., 2021), HallucinationBench (HB) (Guan et al., 2024), MathVista (MV) (Lu et al., 2023), MMBench-TEST-EN (MMB) (Liu et al., 2023c), MMMU-VAL (MMMU) (Yue et al., 2024), MMStar (MS) (Chen et al., 2024a), MMVet (MMV) (Yu et al., 2024), OCRBench (OCRB) (Liu et al., 2023d), ScienceQA (SQA) (Saikh et al., 2022), and MME (Fu et al., 2024) as evaluation metrics. The results of others are sourced from the official OpenCompass publicly available leaderboard (Duan et al., 2024). The best results are **bold**. We provide the specific LLMs employed by each model in Appendix Section D.

| Models | Param (B) | AI2D | HB | MV | MMB | MMMU | MMS | MMV | OCRB | SQA | MME |
|---|---|---|---|---|---|---|---|---|---|---|---|
| *Closed-source VLMs* | | | | | | | | | | | |
| GPT-4o (0513, detail-low) (AI, 2024) | - | 77.4 | 51.7 | 57.2 | 82.8 | 62.8 | 61.6 | 66.5 | 73.5 | 90.1 | 2329 |
| Gemini-1.5-Flash (Gemini Team, 2024) | - | 78.5 | 48.5 | 51.3 | 76.9 | 58.2 | 55.8 | 63.2 | 75.3 | 83.3 | 2078 |
| Claude3.5-Sonnet (Anthropic., 2024) | - | 80.2 | 49.9 | 61.8 | 78.5 | 65.9 | 62.2 | 66.0 | 78.8 | 88.9 | 1920 |
| *Open-source VLMs* | | | | | | | | | | | |
| SmolVLM-500M (Marafioti et al., 2025) | 0.5 | 59.2 | 31.1 | 39.8 | 41.9 | 33.6 | 38.3 | 25.7 | 60.9 | 80.0 | 1395 |
| H2OVL-800M (Galib et al., 2024) | 0.8 | 53.5 | 29.6 | 39.8 | 47.7 | 32.1 | 39.5 | 30.2 | 75.4 | 69.8 | 1469 |
| LLaVA-OV-0.5B (Li et al., 2024a) | 1 | 59.4 | 27.9 | 35.9 | 56.8 | 32.7 | 37.7 | 31.5 | 58.3 | 67.5 | 1449 |
| DeepSeek-VL-1.3B (Lu et al., 2024a) | 2 | 51.5 | 27.6 | 30.7 | 63.8 | 33.8 | 39.9 | 29.2 | 41.3 | 68.4 | 1531 |
| Janus-1.3B (Wu et al., 2024) | 2.1 | 52.8 | 30.3 | 33.7 | 50.3 | 31.2 | 37.6 | 37.5 | 48.2 | 75.1 | 1579 |
| Ovis-1B | 1 | 74.9 | 44.4 | 56.7 | 68.6 | 35.4 | 51.6 | 47.6 | 86.5 | 81.1 | 1664 |
| Ovis-1B + MMCR | | **76.0** | **44.6** | **56.9** | **69.3** | **36.6** | 51.5 | **48.8** | **86.6** | **81.6** | **1689** |
| Phi-3.5-Vision (Marah Abdin, 2024) | 4 | 77.8 | 40.5 | 43.3 | 67.4 | 44.6 | 47.5 | 43.2 | 59.9 | 88.9 | 1838 |
| InternVL-Chat-4B-V1.5 (Chen et al., 2024b) | 4 | 77.0 | 43.0 | 43.6 | 69.7 | 45.1 | 53.1 | 43.6 | 63.9 | 92.6 | 2079 |
| Vintern-3B-beta (Doan et al., 2024) | 3.7 | 69.1 | 43.2 | 43.6 | 66.6 | 46.7 | 47.5 | 37.8 | 61.8 | 75.0 | 1783 |
| XGen-MM-Inst-IL-v1.5 (Le Xue, 2024) | 4.4 | 74.2 | 39.8 | 40.6 | 69.8 | 40.9 | 48.4 | 40.2 | 55.1 | 88.3 | 1809 |
| Ovis-4B | 4 | 84.4 | 52.3 | **66.6** | 78.4 | 49.6 | 59.0 | **62.8** | 87.5 | 92.9 | 2113 |
| Ovis-4B + MMCR | | **84.6** | **52.7** | **66.6** | **79.3** | **50.4** | **59.5** | 62.8 | **87.9** | **93.0** | **2132** |
| Molmo-7B-D (Matt Deitke, 2024) | 8 | 79.6 | 47.7 | 48.7 | 70.9 | 48.7 | 54.4 | 53.3 | 69.4 | 92.2 | 1784 |
| Llama-3.2-11B-VI (Aaron Grattafiori, 2024) | 11 | 77.3 | 40.3 | 47.7 | 65.8 | 48.0 | 49.8 | 57.6 | 75.3 | 83.9 | 1821 |
| Pixtral-12B (Pravesh Agrawal, 2024) | 13 | 79.0 | 47.0 | 56.3 | 72.7 | 51.1 | 54.5 | 58.5 | 68.5 | 87.2 | 1922 |
| WeMM (Jian, 2024) | 7 | 77.9 | 47.5 | 54.9 | 75.7 | 45.3 | 57.0 | 45.0 | 62.8 | 83.3 | 2150 |
| Ovis-8B | 8 | 86.4 | 55.9 | 67.6 | **80.2** | 56.2 | 63.1 | 67.2 | 87.1 | 94.2 | 2164 |
| Ovis-8B + MMCR | | **86.6** | **56.3** | **68.0** | 79.8 | **57.3** | **63.2** | **67.6** | **87.3** | **94.6** | **2237** |

Table 4: Left: Performance of Qwen2.5VL on Public Datasets and MMCR-Bench. Right: Result of Ovis-1B on VisDial val set.

| Qwen2.5VL-3B | AI2D | HB | MV | MMB | MMMU | MMS | MMV | OCRB | MME | PC | CC | LC | CT | DR | OA | | Ovis-1B | Acc. |
|---|---|---|---|---|---|---|---|---|---|---|---|---|---|---|---|---|---|---|
| base | 81.8 | 46.6 | 61.8 | 75.4 | 51.3 | **56.2** | 59.5 | 82.3 | **2149** | 55.2 | 64.5 | 57.0 | 67.5 | 54.6 | 59.7 | | base | 72.8 |
| + MMCR | **82.1** | **47.9** | **61.9** | **76.1** | **52.9** | 55.9 | **60.8** | **82.9** | 2141 | **58.9** | **66.6** | **60.9** | **70.2** | **62.9** | **63.0** | | + MMCR | **75.1** |

pronounced in smaller models. Moreover, significant score enhancements in the corresponding metric post fine-tuning prove that our MMCR instruction fine-tuning dataset is highly effective in mitigating model redundancy.

## 4.2 COMPARISON AND IMPROVEMENT ON SOTA

In Tab. 3, we present the performance of a range of released open-source and closed-source models across multiple public evaluation benchmarks. We group the open-source models and the various-sized Ovis models by their parameter counts into three groups, and incorporate MMCR data into the instruction fine-tuning phase of the Ovis models. This setup not only demonstrates the powerful multimodal capabilities of our baseline Ovis model, but also shows that our proposed MMCR instruction fine-tuning data yields considerable performance improvements. The results indicate that MMCR data can bring noticeable enhancements on existing public benchmarks. For instance, with the Ovis-1B model, the introduction of MMCR leads to improvements of +1.1% on AI2D, +1.2% on MMMU and MMVet, and an overall boost of 0.6% across a total of 10 datasets. These enhancements underscore the contribution of our MMCR dataset to the performance of VLMs on general multimodal tasks. We additionally select Qwen2.5VL (Bai et al., 2025) to validate MMCR. As shown in Tab. 4 Left, where MMCR still brings objective performance gains.

## 4.3 APPLES-TO-APPLES COMPARISON

To further demonstrate the superior quality of our proposed MMCR dataset, we conduct a performance comparison on the same baseline with another same type dataset MMDU (Liu et al., 2024c) under approximately equal data volumes (50k for MMCR, 45k for MMDU). As shown in Tab. 5, both MMDU and MMCR improve performance compared to the baseline, and our proposed MMCR delivers even greater gains, further validating the quality of MMCR. Besides, We opt for VisDial (Das et al., 2017), a similarly multi-turn dialogue dataset, to demonstrate the utility of

Table 5: Comparison with baseline and MMDU on Ovis-1B. The best results are **bold** and the second-best results are underlined.

| Model Ovis-1B | AI2D | HB | MV | MMB | MMMU |
|---|---|---|---|---|---|
| base | 74.9 | 44.4 | 56.7 | 68.6 | 35.4 |
| MMDU | 75.1 | 44.2 | 55.9 | 67.8 | 35.9 |
| MMCR | **76.0** | **44.6** | **56.9** | **69.3** | **36.6** |

| Model Ovis-1B | MMS | MMV | OCRB | SQA | MME |
|---|---|---|---|---|---|
| base | **51.6** | 47.6 | 86.5 | 81.1 | 1664 |
| MMDU | 51.3 | **50.9** | 86.6 | 81.5 | 1667 |
| MMCR | 51.5 | 48.8 | 86.6 | 81.6 | **1689** |

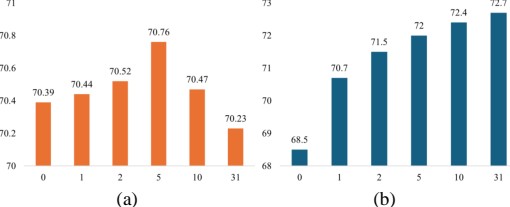

Figure 5: Histogram of Data Volume vs. Performance. (a) Average score across 9 benchmarks for Ovis-4B. (b) OA score on MMCR-Bench for Ovis-4B. X-axis: data volumes (unit: 10k).

MMCR in enhancing multi-turn conversational capabilities within the model. We use its val set, where model receives first nine turns of dialogue and image, then outputs response for last turn. Response was delivered to 4o along with ground truth, and evaluation was conducted solely from Accuracy dimension. As shown in Tab. 4 Right, this resulted in a 2.3% improvement across 2064 samples, demonstrating that MMCR contributes to model's multi-turn dialogue capabilities.

### 4.4 BIASED PHENOMENON: LESS IS MORE

Fine-tuning phase is crucial for VLMs' response performance. Although more data is commonly believed to enhance results, leading researchers to use ever-larger datasets, an often overlooked factor is the importance of broad coverage and balanced distribution when designing fine-tuning datasets. As shown in Fig. 5, our experiments indicate that increasing MMCR data during single instruction fine-tuning initially boosts average performance on both public benchmarks and MMCR-Bench. However, once MMCR data exceeds a certain proportion, public benchmark performance declines even as MMCR-Bench results continue to improve.

We clarify that performance decline is not due to data overlap, but rather result of task-specificity. We use OmniCorpus as data sources, which covers diverse websites, along with prompt engineering to ensure that the data strictly adhere to the visual information, eliminating any potential creative deviations. This approach guarantees the diversity and quality of MMCR. MMCR is designed for multi-turn dialogue, the capacity behind this is different from that for existing benchmarks. Properly enhancing multi-turn dialogue ability can help other perception ability, while singly improving this ability too much may lead to ability bias, which affects performance on common existing benchmarks. It should be emphasized that our post-training phase utilizing MMCR incorporates only approximately 250k samples, as an excessive proportion will result in the aforementioned phenomenon. We confirm that for comprehensive perception capabilities, it is necessary to fine-tune VLM using different types of datasets with proper ratio. The multistage, multidata source, and multitask training strategy employed in mainstream works also supports our viewpoint.

## 5 CONCLUSION

In this paper, to enhance VLMs to meet the needs of daily human-AI conversations, we present MMCR, a multimodal multi-turn dialogue dataset. We thoroughly consider the characteristics of daily human conversations, such as multi-turn dialogues typically revolving around a few central topics, contextual referencing to maintain logical consistency. Through prompt engineering, we infuse this information into GPT-4o to generate the data. Using the CLIP model, we filter 1.6 million samples to obtain 310k high-quality single/multi-image, multi-turn dialogue samples. Experimental validation on the recently open-sourced Ovis show that fine-tuning with the proposed MMCR not only improves the metrics on MMCR-Bench but also further enhances the model's performance on public benchmarks. Finally, based on our experiments, we highlight a point often overlooked in training large models: more fine-tuning data is not always better, and maintaining a balanced proportion of data across different tasks is equally important.

## 6    REPRODUCIBILITY STATEMENT

To ensure reproducibility of our results, we provide the experimental settings in Appendix Section B, as well as the generated prompts in Section E and the judgment prompts in Section F.

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

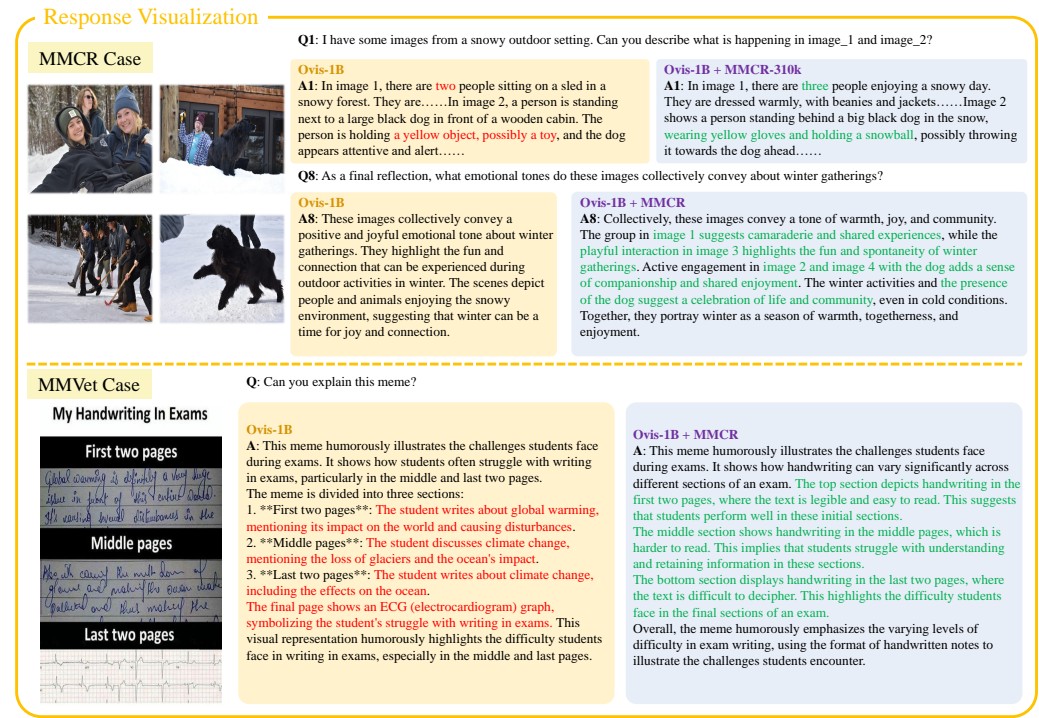

Figure 6: Visualization of Response Comparison. We present a comparison between the baseline Ovis-1B and the model fine-tuned with MMCR data. Errors and hallucinations are marked in red, while detailed and accurate descriptions are marked in green.

# A   LARGE LANGUAGE MODELS USAGE STATEMENT

Large Language Models were only used to aid in polishing the manuscript text.

# B   IMPLEMENTATION DETAILS

We use Ovis as the baseline model for all experiments. Ovis encompasses three configurations: the LLM module, the ViT backbone, and the visual vocabulary size. We incorporate popular open-source LLM (Qwen2.5-Instruct Qwen et al. (2025) and ViT (aimv2 Fini et al. (2024b)) into Ovis. The size of the visual vocabulary is set to 65536.

We strictly adhere to the original Ovis settings to ensure fairness. The batch size is 1024, with 1 epoch. The experiments use a cosine annealing learning rate of 2e-6 and a weight decay of 0. All experiments are conducted using the PyTorch framework on 8 H100 80G GPUs. Evaluation of public datasets is based on VLMEvalKit, while evaluation of MMCR Bench is conducted by comparing model's responses with ground truth using 4o. For multi-turn dialogue data, model will reference previous turns of the dialogue each time it responds to current question, not simultaneously or only last one.

Regarding the number of images adopted and dialogue turns, our experimental findings demonstrate that 4 rounds of dialogue per single image can adequately describe the visual content without causing the model to generate creative content that deviates from the image, which would introduce hallucination problems during the training process. Additionally, to maintain consistent batch size and other configurations with the pre-training phase while avoiding out-of-memory issues, we opted for 2-4 images with dialogue sequences extending up to 8 turns.

| Models | LLM |
|---|---|
| *Closed-source VLMs* | |
| GPT-4o (0513, detail-low) | - |
| Gemini-1.5-Flash | - |
| Claude3.5-Sonnet | - |
| *Open-source VLMs* | |
| SmolVLM-500M | SmolLM2-360M |
| H2OVL-800M | H2O-DANUBE3 |
| LLaVA-OV-0.5B | Qwen2-0.5B |
| DeepSeek-VL-1.3B | DeekSeek-1B |
| Janus-1.3B | DeepSeek-1B |
| Phi-3.5-Vision | Phi-3.5 |
| InternVL-Chat-4B-V1.5 | Phi-3 |
| Vintern-3B-beta | Qwen-2.5-3B |
| XGen-MM-Inst-IL-v1.5 | Phi-3 |
| Molmo-7B-D | Qwen2-7B |
| Llama-3.2-11B-VI | Llama-3.1-8B |
| Pixtral-12B | Nemo-12B |
| WeMM | InternLM2-7B |

Table 6: Models and their corresponding LLM backbones.

## C  RESPONSE VISUALIZATION

We visually demonstrate the error cases and the optimizations brought by MMCR through model responses. Fig. 6 shows a comparison between the baseline and MMCR-enhanced model responses on two evaluation datasets—MMCR-Bench and MMVet. It is evident that after incorporating MMCR data, the model's responses become more accurate and clear, effectively mitigating issues of hallucination and vague understanding. The improved responses reveal a deeper comprehension of the underlying meanings in the images, rather than merely interpreting their symbolic shapes.

## D  UTILIZATION OF LLMs

Tab. 6 presents the correspondence between various VLM models in our experiments and the underlying LLMs they employ.

## E  GENERATE PROMPT

We have meticulously designed a prompt for GPT-4o to generate multi-image, multi-turn dialogues with strong contextual relevance, thereby more efficiently utilizing the provided images and text. The exact prompt is shown in Fig. 7. First, we define the task: the agent must simulate a real-world scenario where a human provides images, background information, and their own requests in an interaction with an intelligent assistant. The key requirement is that the overall dialogue maintains clear contextual continuity, with subsequent questions based on previous requests or questions and their corresponding answers, which aligns with the original intention behind designing MMCR. Second, to ensure that the generated data can be effectively used for the instruction fine-tuning phase of VLMs, we emphasize that only the human should introduce images, and images with significantly disparate content should be distributed across multiple dialogue turns to avoid abrupt transitions. Since the accuracy of the assistant's responses—ensuring that they are consistent with the provided data and free from hallucinations or creative interpretations—is crucial for the dataset's quality, we require that the generated dialogues adhere strictly to the provided data without introducing fictional content. Next, we stress the importance of adhering to a standardized data format and avoiding any content that may contain harmful elements. Finally, to ensure the overall quality of the generated data, we instruct GPT-4o to recheck the generated multimodal, multi-turn dialogues against all the above requirements. This carefully crafted prompt has proven to be highly effective in guiding the generation process.

# F  JUDGMENT PROMPT

In Fig. 8, we present the prompt used to evaluate the model responses against the reference provided in MMCR-Bench. This prompt guides GPT-4o to assess the responses from six aspects: Precision, Consistency, Logical, Clarity, Redundancy, and Overall Score. Each dimension is divided into five scoring intervals, with corresponding descriptions defined for each interval to enable accurate judgment of the model's performance. Finally, we aggregate the scores from all 600 evaluation samples in MMCR-Bench by summing them, dividing by the total number of samples, and then multiplying by 10 to obtain the final score for the model's responses.

# G  DATA EXAMPLE

In Fig. 9 and Fig. 10, we present two multimodal multi-turn dialogue samples from MMCR-310k and MMCR-Bench, respectively, to illustrate the characteristics of our data. Overall, our data features long-duration multi-turn dialogues, involves the understanding and perception of multiple images, maintains rigorous contextual logical relationships, and consistently focuses on discussions around specific images with a clearly defined theme.

Specifically, we use "Image_n: ⟨image⟩," in the dialogue to specify the position for the corresponding image token, where n denotes the specific image number. In the example shown in Fig. 9, the human's Question 1 initially involves two images. After prompting the intelligent assistant for descriptions, the conversation expands into a discussion about the lifestyles represented by each image. Subsequently, in Question 3, the third image is introduced, leading to a discussion of the relationships between Image 2 and Image 3, as well as between Image 1 and Image 3, and culminating in a comprehensive discussion that summarizes all the images. This setup challenges the VLM's ability to understand and retain long texts. Similarly, in the example from Fig. 10, the human's question involves multiple images and ultimately revolves around the central dialogue theme. These examples not only highlight the challenges of long-text understanding and memory for VLMs but also provide a comprehensive evaluation of the model's contextual reasoning and multi-image perceptual understanding capabilities.

# Rule settings:

You are an expert in generating multi-round conversations between a Human and a chatbot Assistant based on provided images and interleaved text.

### Given the images and interleaved text, your task is to:

**1.** Create a multi-round conversation between a Human and an Assistant based on the provided images. The dialogue simulates real-world scenarios where a Human makes requests or asks questions, and the Assistant provides responses. Ensure that each round of conversation gradually deepens the exploration of the image details, connections between images, and relevant themes. The conversation must maintain a clear contextual connection, with follow-up questions building on the previous requests or questions and their corresponding responses. The total number of conversation turns MUST be at least eight rounds.

**2.** In the dialogue, ONLY the Human introduces the images, and only the first time an image is mentioned, it should be referenced using <image n>. In all subsequent conversations, refer to it as image n. Suggest putting similar images in once Human's requests/questions; if the images have significantly different content, they should appear in different dialogue turns to ensure logical coherence in the context. NOTE: The Assistant's responses MUST not omit any of the images mentioned by humans in the corresponding turns.

**3.** The Human's messages must include references to all provided images, ensuring that each image is referenced exactly once in the Human's requests/questions. The conversation should be designed so that all images are naturally incorporated into the dialogue.

**4.** The Assistant's responses MUST cover all the images mentioned in the human's requests/questions and be detailed and contextually relevant, focusing solely on observable details directly inferred from the images. Avoid describing imaginary content or introducing creative interpretations. Do not describe the contents by itemizing them in list form, and minimize aesthetic descriptions as much as possible. The responses must build upon previous discussions and provide continuity, focusing only on what is confidently observable in the images. When referencing images, the Assistant MUST uses image n (without angle brackets) in their responses. Avoid unnecessary repetition.

**5.** Each turn in the conversation must follow this format:
    < Human $m$ >: [Human's requests/questions that include '<image n>' or 'image n']
    < Assistant $m$ >: [Assistant's response]
where m denotes the $m^{th}$ round of dialogue.

**6.** For any unreasonable or inappropriate requests from the Human (those that are harmful to society, immoral, or illegal), the Assistant should politely refuse, explaining the reasoning clearly and empathetically, and offering guidance or safe alternatives where appropriate.

**7.** Before concluding the conversation, please carefully verify that all image references correctly correspond to the provided images, ensure that the numbering is accurate, that responses align with the image content, and that each of the assistant's answers includes every image mentioned by humans in their respective turns without omission.

### You need to generate conversations based on the image and interleaved text below:
Image:
{[image list]}

Interleaved text:

{[text]}

### Here is an example of generating multi-round conversations:

{Example}

Figure 7: Dialogue generation prompt.

You are an assistant skilled at evaluating the performance of multi-modal models in multi-turn dialogues.

Please act as an impartial judge and evaluate the overall performance of the interaction, which includes both image descriptions and dialogue exchanges.

We will provide you with model outputs and reference examples for your evaluation.

You will need to assess the performance based on the following scoring rules:

1. Evaluate each dimension, pointing out the strengths and weaknesses, and assign a score between 1 and 10 for each.

2. Consider the precision and conciseness of image descriptions, the consistency of contextual references, the logical progression between dialogue turns, the clarity of the dialogue theme, and the absence of redundant content.

3. Based on your evaluations for each individual aspect, provide an overall score for the entire interaction on a scale of 1 to 10.

4. Automatic Penalty Triggers:

- Length penalty: If the length differs from the reference answer by more than 20%, 1 point will be deducted from Precision.

- Repeated punishment: For each repetition of previous information, 1 point will be deducted from Redundancy.

- Speculation Penalty: Unsubstantiated or speculative claims will deduct 2 points from Logical.

### Scoring rules:

Precision and Conciseness of Image Descriptions
Scores 1-2: Descriptions are very vague, excessively wordy, or missing critical visual details.
Scores 3-4: Descriptions provide some accurate information but include unnecessary or unclear details.
Scores 5-6: Descriptions are adequate and mostly clear, covering the essential visual elements without being outstanding.
Scores 7-8: Descriptions are precise and concise, effectively highlighting important visual features; however, minor issues may remain.
Scores 9-10: Descriptions are exceptionally precise, concise, and insightful with flawless detail.

Consistency of Contextual References
Scores 1-2: Contextual references are confusing, inconsistent, or largely misaligned with the dialogue.
Scores 3-4: References are somewhat ambiguous or uneven, leading to occasional lapses in clarity.
Scores 5-6: References are generally clear and consistent with only minor lapses that do not majorly disrupt understanding.
Scores 7-8: References are clearly maintained and consistent, linking well to previous content with only rare exceptions.
Scores 9-10: References are perfectly clear and consistent, leaving no room for ambiguity.

Logical Contextual Relationship
Scores 1-2: The dialogue shows a lack of logical flow with disjointed or disconnected turns.
Scores 3-4: There are noticeable gaps or weak logical connections between turns.
Scores 5-6: The dialogue maintains a basic logical flow with some minor leaps or less-than-smooth transitions.
Scores 7-8: The dialogue flows logically and coherently with smooth transitions, though not without minor imperfections.
Scores 9-10: The dialogue is flawlessly logical, with seamless and well-reasoned transitions throughout.

Clarity of Dialogue Theme
Scores 1-2: The dialogue theme is unclear, inconsistent, or frequently shifts, resulting in confusion.
Scores 3-4: The theme is only partially evident and sometimes lacks focus or consistency.
Scores 5-6: The theme is generally clear and maintained, though occasional deviations occur.
Scores 7-8: The dialogue maintains a clear and consistent theme throughout with only slight lapses.
Scores 9-10: The theme is exceptionally clear, well-defined, and consistently adhered to from beginning to end.

Absence of Redundancy
Scores 1-2: The dialogue contains significant repetition or redundant content that disrupts the flow.
Scores 3-4: There is noticeable redundancy that sometimes affects the overall fluidity of the dialogue.
Scores 5-6: Minor redundancy is present, but it does not significantly hinder the interaction.
Scores 7-8: The dialogue is largely free of repetition, with smooth and natural transitions; only occasional redundancy is observed.
Scores 9-10: The dialogue is completely devoid of redundancy, with elegantly seamless transitions.

Overall Score:
Scores 1-2: The performance is extremely poor, with major issues across all dimensions.
Scores 3-4: The performance is below average, with several critical shortcomings.
Scores 5-6: The performance meets basic requirements and is acceptable, though there is noticeable room for improvement.
Scores 7-8: The performance is strong and effectively meets most evaluation criteria, with only minor imperfections.
Scores 9-10: The performance is outstanding, fully addressing all dimensions with exceptional quality.

### Output Format

Please remember, you must evaluate and explain each dimension before scoring. After your explanation for each dimension, add the score for that dimension. Finally, at the end of your response, in the format of a dictionary (including brackets), return all your scoring results, ensuring your scores are integers:

{'Precision': Score, 'Consistency': Score, 'Logical': Score, 'Clarity': Score, 'Redundancy': Score, 'Overall Score': Score}

For example: {'Precision': 5, 'Consistency': 6, 'Logical': 5, 'Clarity': 6, 'Redundancy': 7, 'Overall Score': 6}

Figure 8: Judgment prompt for GPT-4o.

## Dialogue Samples

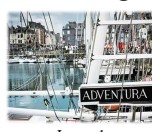 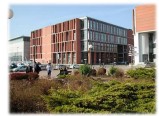

Image 1       Image 2

### Question1:

Image_1: <image>\nImage_2: <image>\nI've got a couple of interesting images. Can you describe what's going on in both image 1 and image 2?

### Assistant1:

Image 1 features a marina scene with a sailboat prominently displaying the name "ADVENTURA." The background shows a charming harbor town with traditional architecture and boats docked along the water, creating a vibrant, maritime atmosphere. Image 2, on the other hand, depicts a modern building with a brick and large glass windows. This building sits in an urban setting with cars parked outside and a few people walking nearby, contrasting the rustic, nautical theme of image 1 with a more contemporary, city-like environment.

### Question2:

The contrast between the harbor scene in image 1 and the urban setting in image 2 is interesting. Can you elaborate on how these two images might reflect different lifestyles or environments?

### Assistant2:

Certainly! Image 1 invites us into a lifestyle closely tied to the sea, offering a sense of adventure and exploration often associated with sailing. The maritime town and traditional architecture emphasize a textured, historical ambiance. In contrast, image 2 represents an urban lifestyle, characterized by modern buildings and a bustling environment. This setting reflects efficiency, progress, and connectivity, often found in cities. Together, these images capture two very distinct environments one that embraces tradition and nature, and another that leans into modernity and human innovation.

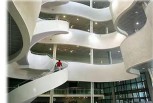

Image 3

### Question3:

Image_3: <image>\nThat's a fascinating take! Now, how does image 3 fit into this discussion, considering the architectural aspect?

### Assistant3:

Image 3 features an interior view with striking architectural design, showcasing sweeping, curved white staircases that create a sense of fluid motion. This modern interior aligns more closely with the urban theme in image 2 due to its contemporary design elements and innovative use of space. The emphasis on curves and open space contrasts with the straightforward, linear architecture typically found in traditional settings like the one in image 1. Image 3 highlights creativity and modern aesthetics in architecture, illustrating the transformative potential of interior design.

### Question4:

So image 3 continues this modern theme. What relationship do you see between the architectural design in image 3 and the exterior view shown in image 2?

### Assistant4:

The architectural design in image 3 complements the modern building depicted in image 2 through its emphasis on innovation and style. Both images suggest a preference for modernism, with image 2 showcasing this through its exterior brick and glass, and image 3 through its sleek, flowing interior design. The interplay between exterior and interior architecture captures the essence of contemporary spaces both functionally and aesthetically creating continuity between how a structure looks from the outside and how it feels inside.

### Question5:

I see how image 3 ties into the urban theme. How about the cultural or historical significance? Could you identify any elements in image 1 that might touch upon these aspects?

### Assistant5:

Image 1, with its harbor scene, captures cultural and historical elements through its depiction of the maritime lifestyle. The traditional architecture of the harbor town reflects historical significance, evoking images of centuries-old coastal communities reliant on fishing and sea trade. The presence of the sailboat named "ADVENTURA" emphasizes this adventurous and exploratory spirit, suggesting a connection to historical seafaring journeys. Together, these elements convey a deep cultural connection to the sea and a way of life rooted in tradition and exploration.

### Question6:

Speaking of tradition and modernity, how do you think the contrast between the traditional elements in image 1 and the contemporary aspects in images 2 and 3 might reflect on societal change?

### Assistant6:

The contrast between image 1 and images 2 and 3 highlights the ongoing societal transition from traditional lifestyles to modern urban living. Image 1 embodies longstanding cultural practices and a lifestyle closely tied to nature and historical roots. Meanwhile, images 2 and 3 represent the shift towards modernity, showcasing urbanization, architectural innovation, and a focus on design and functionality. This evolution reflects broader societal changes where technology and urban living increasingly shape our daily experiences, contrasting with heritage and conventional ways of life.

### Question7:

That's insightful! What about the emotions or lifestyle aspirations these images might invoke?

### Assistant7:

The images conjure distinct lifestyle aspirations: Image 1 may invoke a longing for simplicity, adventure, and a deeper connection to nature and history, with aspirations of fulfilling a dream to explore or live by the sea. Conversely, images 2 and 3 lean toward aspirations associated with modernity such as efficiency, innovation, and the conveniences of urban living. They might appeal to those who value contemporary aesthetics, advanced architecture, and the connectivity of city life. Each image taps into different desires and inspirations, reflecting varied emotional landscapes.

### Question8:

Finally, do you think these images, when considered together, tell a broader story about diversity in how we experience the world?

### Assistant8:

Absolutely, when viewed together, these images create a narrative about the diversity in human experiences and environments. Image 1 embodies a timeless connection to nature and tradition, while images 2 and 3 reflect the constantly evolving landscape of urban development and innovation. Together, they illustrate the multifaceted nature of human existence how we navigate the balance between honoring tradition and embracing modernity. This diversity enriches our collective experience, showcasing the varied paths through which people find fulfillment and meaning.

Figure 9: MMCR-310k Example.

## Dialogue Samples

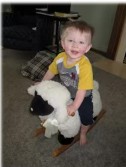

Image 1          Image 2

### Question1:
Image_1: <image>\nImage_2: <image>\nI have some images showing kids with a toy. Could you describe what's happening in image_1 and image_2?

### Assistant1:
In image 1, a child dressed in a white sweater is interacting with a large rocking sheep toy in a room. The child appears to be focused on the toy, which adds a sense of curiosity and exploration. Image 2 depicts a child sitting on the same rocking sheep toy, wearing a yellow and gray outfit, in what seems to be a living room setting. The child looks engaged in play, suggesting a moment of joy and imagination.

### Question2:
Interesting! How does the playfulness in these images resonate with each other?

### Assistant2:
Both images capture the children's playful interactions with the rocking sheep toy. In image 1, the child's focus on the toy suggests an explorative play, while in image 2, the engagement of sitting atop the toy highlights imaginative play, perhaps pretending to ride the sheep. These interactions reflect different facets of how children explore and enjoy their toys, with both scenes indicating a sense of wonder and creativity.

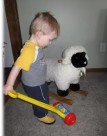

Image 3

### Question3:
Image_3: <image>\nGreat observations. Now, what about image_3? How does the activity in that image compare to the others?

### Assistant3:
Image 3 shows a child walking while holding a toy hammer that is brightly colored. The rocking sheep toy is also present in the scene, indicating that the child might be transitioning between different play activities. Compared to the seated play in image 2, image 3 suggests a more active and dynamic form of play, with the child exploring the space and engaging with multiple toys, which can broaden their imaginative and physical interactions.

### Question4:
How do you think the presence of the same toy affects the theme of play in these images?

### Assistant4:
The recurring presence of the rocking sheep toy in all images provides a continuity that ties the different scenes of play together. It serves as a central object around which various forms of play revolve standing and observing, sitting and riding, and actively moving around it. This continuity emphasizes how a single toy can foster diverse play experiences, from imaginative scenarios to physical engagement, highlighting the adaptability and creativity in a child's playtime.

### Question5:
It's fascinating how versatile the play can be. Focusing on the environment, does the setting influence the interaction in these images?

### Assistant5:
Yes, the settings in the images do influence the interactions. Image 1, taking place in a room, suggests a spontaneous encounter with the toy, possibly during family routines. Image 2's living room setting creates a cozy space for imaginative play, where the child feels at home. Image 3, still in the living room, allows for more active movement and varied play, demonstrating how the environment can shape the manner and style of interaction, supporting different play activities and moods.

### Question6:
That makes a lot of sense. Now, thinking about the expressions and postures in image 2 and image 3, what can we infer about the child's engagement level?

### Assistant6:
The child in image 2, while sitting on the rocking sheep, appears to be engaged in a focused and immersive form of play, likely imagining riding or interacting with the toy in a meaningful way. In contrast, in image 3, the child holding a toy hammer seems to be in a more explorative and active mode. This suggests high engagement levels but different styles immersive and imaginative in image 2, and dynamic and experimental in image 3.

### Question7:
It's wonderful to see those interactions. Lastly, how do these playful moments contribute to a child's development?

### Assistant7:
These playful interactions contribute significantly to a child's development. In all images, the child is exploring different aspects of play: cognitive and imaginative development through engaging with the rocking sheep and physical and motor skills through movement and tool use in image 3. These activities not only foster creativity and imagination but also encourage motor skill refinement, problem-solving, and emotional expression, all crucial for holistic development.

### Question8:
Play certainly seems important! Thank you for explaining how these images connect and contribute to growth.

### Assistant8:
You're welcome! Play is indeed a vital part of childhood, offering a foundation for many developmental milestones. If you have more questions or need further analysis, feel free to ask!

Figure 10: MMCR-Bench Example.

