# OpenReview forum: "MMCR: Advancing Visual Language Model in Multimodal Multi-Turn Contextual Reasoning"
_ICLR.cc/2026/Conference — ICLR 2026 Conference Withdrawn Submission_

### Official Review · Reviewer_u6BU · 2025-10-26

**Soundness:** 1
**Presentation:** 2
**Contribution:** 2
**Rating:** 2
**Confidence:** 4

**Summary:**

This paper presents MMCR, a dataset and benchmark derived from OmniCorpus to enhance the multi-turn, multi-image dialogue capabilities of vision–language models. It includes MMCR-310k, the largest instruction-tuning dataset with 310K dialogues (covering 1–4 images and 4 or 8 turns), and MMCR-Bench, a diagnostic benchmark spanning multiple domains with both CLIP-based semantic filtering and manual review. Fine-tuning multiple models on MMCR-310k results in consistent performance improvements on MMCR-Bench.

**Strengths:**

This work proposed a well-designed dataset that addresses the need for multi-image, multi-turn dialogue data. The dataset shows strong contextual relevance and logical progression across dialogue turns. The MMCR-Bench benefits from rigorous CLIP-based filtering and, importantly, multi-expert human supervision, ensuring high data quality. Experiments show that fine-tuning on MMCR-310k consistently improves performance across three model scales and multiple public benchmarks, reinforcing the practical value of AI-generated synthetic data for enhancing vision–language models.

**Weaknesses:**

- In Section 4.3, the claim that MMCR-50k and MMDU-45k are “approximately equal” is questionable, as MMCR-50k has about 22% more data. It is unclear whether the improvement comes from dataset scaling or from MMCR’s design itself.
- In Section 4.4, the discussion on redundancy is confusing. The first paragraph promotes using ever-larger datasets, while the second emphasises combining different dataset types with proper ratios. These views are not contradictory (they differ only in scope), so presenting this as evidence for the “less-is-more” effect is inaccurate.
- If the conclusion in Section 4.4 (Figure 5) holds, then the comparison in Section 4.3 becomes unfair, as MMCR-50k would merely coincide with the model’s performance peak (across benchmarks) when trained solely on MMCR data. Moreover, this interpretation contradicts
- Section 4.2 (Table 3), where fine-tuning on MMCR-310k leads to consistent gains across Ovis-1B/4B/8B and multiple benchmarks. If “less-is-more” is true, performance with the full 310 k dataset should have been lower, not higher.

**Questions:**

- In Section 4.3, what are the results when using a 45k volume for MMCR to match MMDU?
- Sections 4.3 and 4.4 use different base models. Would the findings remain consistent if both experiments used the same model scale (e.g., 1B or 4B)?
- If the volume of other datasets were increased proportionally, would the trend in Figure 5 still hold, or is it simply overfitting within a single dataset rather than a general principle?
- Why does Table 3, where models fine-tuned on MMCR-310k show gains, contradict Figure 5, which suggests performance degradation with larger data volumes?

---

### Official Review · Reviewer_PPNV · 2025-11-01

**Soundness:** 2
**Presentation:** 3
**Contribution:** 2
**Rating:** 4
**Confidence:** 4

**Summary:**

This paper introduces MMCR, a large-scale dataset and benchmark designed for multimodal multi-turn dialogues. It uses GPT-4o to synthesize dialogue data from OmniCorpus and uses CLIP to strictly filter  the synthesized samples. Additional human supervision is designed to control the quality of MMCR-Bench. Experiments demonstrate improvements on both MMCR-Bench and general multimodal benchmarks when fine-tuned with MMCR.

**Strengths:**

* This work contribute a new dataset addressing multimodal multi-turn dialogues.

* The paper is well-structured and easy to follow.

* The proposed data curatoin method is reasonable.

* Strong empirical evidence supporting dataset usefulness across VLM sizes (1B–8B).

**Weaknesses:**

* The proposed method heavily depends on closed-source models (GPT-4o) for both data generation and judging. This raise a big concern: would the dataset actually is a distillation from closed-source models?

**Questions:**

* Since my biggest concern comes from the same model (GPT-4o) as both data generator and the judge, what would happen if you use GPT-4o as data generator, but another model as judge?

---

### Official Review · Reviewer_cBt2 · 2025-11-03

**Soundness:** 3
**Presentation:** 4
**Contribution:** 3
**Rating:** 4
**Confidence:** 4

**Summary:**

This paper proposes a novel dataset MMCR comprises of MMCR-310k with 310K contextual dialogues each covering 1-4 images and 4 or 8 dialogue turns, and MMCR-Bench, a diagnostic benchmark featuring single/multi-image mixed dialogues, spanning 8 domains and 40 sub-topics. The evaluation results show that models fine-tuned with MMCR-310k achieve 5.2% higher contextual accuracy on MMCR-Bench and achieve improvements on existing multimodal benchmarks such as AI2D, MMMU and MMVet. In addition, this paper discover the “Less is More” phenomenon in supervised fine-tuning.

**Strengths:**

1. The proposed MMCR is the largest multi-turn instruction tuning dataset with 310K contextual dialogues for training and 600 dialogues for evaluation, each covering 1-4 images and 4 or 8 dialogue turns.
2. The experimental results show that models fine-tuned with MMCR-310k achieve higher contextual accuracy on MMCR-Bench as well as existing multimodal benchmarks.

**Weaknesses:**

1. For evaluation, Consistency score and Logical score are used to judge the multi-turn dialogues ability of model on MMCR-Bench. However, when building MMCR-310K, how to evaluation the Consistency score and Logical score of generated training samples is not mentioned in this paper, while only image-text similarity is used to filter samples in generation pipeline. Therefore, I am concerned about the logical consistency of the generated multi-turn dialogue samples and the diversity of logical relationships within multi-turn dialogues for MMCR-310K.
2. It is better to show the data sample distribution over 8 domains and 40 sub-topics of both MMCR-310K and MMCR-Bench.
3. This paper would benefit from showing more qualitative examples of MMCR-Bench inferred by the models.

**Questions:**

1. For the experiment on Section 4.4, the model trained with 5k samples of MMCR achieves highest benchmark performance. Whether different proportions randomly sampled from MMCR used for training can obtain the same conclusion?
2. Can the LESS IS MORE phenomenon also observed in Ovis-1B and Ovis-8B?

---

### Note · Authors · 2025-11-20

I have read and agree with the venue's withdrawal policy on behalf of myself and my co-authors.